# The Impact of the FilmArray-Based Detection of Microbial Pathogens from Positive Blood Culture Vials on the Time to Optimal Antimicrobial Regimen in Intensive Care Units of the Helios University Clinic Wuppertal, Germany

**DOI:** 10.3390/jcm10245880

**Published:** 2021-12-15

**Authors:** Jannik Schumann, Ulrike Johanns, Parviz Ahmad-Nejad, Beniam Ghebremedhin, Gabriele Woebker

**Affiliations:** 1Center for Clinical and Translational Research (CCTR), Institute for Medical Laboratory Diagnostics, Helios University Hospital Wuppertal, Witten/Herdecke University, 42283 Wuppertal, Germany; jannik.schumann@uni-wh.de (J.S.); parviz.ahmad-nejad@helios-gesundheit.de (P.A.-N.); beniam.ghebremedhin@uni-wh.de (B.G.); 2Clinic for Intensive Care Medicine, Helios University Hospital Wuppertal, Witten/Herdecke University, 42283 Wuppertal, Germany; ulrike.johanns@helios-gesundheit.de

**Keywords:** sepsis, septic shock, bloodstream infections, ICU, blood culture, RT PCR, identification of pathogens, antibiotic resistance, length of stay, ventilation duration, appropriate antimicrobial regimen

## Abstract

The role of empirical therapy and time to first effective treatment, including the antimicrobial stewardship program, are decisive in patients presenting with bloodstream infections (BSI). The FilmArray^®^ Blood Culture Identification Panel (FA BCID 1.0) detects 24 bacterial and fungal pathogens as well as 3 resistance genes from positive blood cultures in approximately 70 min. In this paper, we evaluate the impact of the additional FA BCID analysis on the time to an optimal antimicrobial therapy and on the length of stay in the ICU, ICU mortality, and PCT level reduction. This retro-/prospective trial was conducted in BSI patients in the ICU at a German tertiary care hospital. A total of 179 individual patients with 200 episodes of BSI were included in the prospective intervention group, and 150 patients with 170 episodes of BSI in the retrospective control group. In the intervention group, BSI data were analyzed including the MALDI-TOF MS (matrix assisted laser desorption ionization time-of-flight mass spectrometry) and FA BCID results from January 2019 to August 2020; the data from the control group, including the MALDI-TOF results, were collected retrospectively from the year 2018. The effective and appropriate antimicrobial regimen occurred in a median of 17 hours earlier in the intervention versus control group (*p* = 0.071). Furthermore, changes in the antimicrobial regimens of the intervention group that did not immediately lead to an optimal therapy occurred significantly earlier by a median of 24 hours (*p* = 0.029). Surrogate markers, indicating an earlier recovery of the patients from the intervention group, such as length of stay at the ICU, duration of mechanical ventilation, or an earlier reduction in PCT level, were not significantly affected. However, mortality did not differ between the patient groups. A postulated reduction of the antimicrobial therapy, in those cases in which coagulase-negative *Staphylococcus* species were identified, did occur in the control group, but not in the intervention group (*p* = 0.041). The implementation of FA BCID into the laboratory workflow can improve patient care by optimizing antimicrobial regimen earlier in BSI patients as it provides rapid and accurate results for key pathogens associated with BSI, as well as important antimicrobial resistance markers, e.g., *mecA* or *vanA*.

## 1. Introduction

Sepsis and bloodstream infections (BSI) still pose diagnostic and therapeutic challenges. BSI are among the leading causes of morbidity and mortality worldwide. An estimated 19 million hospitalizations in intensive care units occur worldwide each year, resulting in a death rate of 25% in these patients [1,2,3]. A timely administration of antimicrobial therapy reduces mortality and morbidity, as long as it is effective against the causative microorganism of the BSI or sepsis [3,4,5,6]. With the wide availability of broad-spectrum antimicrobials, administering an effective and appropriate antimicrobial regimen at the initial phase has been related to a better outcome, especially for patients with severe sepsis or septic shock. However, the empiric administration of broad-spectrum antibiotics leads to unnecessary antibiotic exposure and to a high risk of selecting drug-resistant microorganisms.

Because the long-term use of broad-spectrum antimicrobials is associated with its own complications, a switch to narrow-spectrum antimicrobials has proven to be a valuable therapeutic step [3,7,8]. The administration of such a small spectrum antimicrobial requires a much more detailed knowledge about the causative microorganism. Today’s gold standard in identifying these causative microorganisms is the drawing of blood cultures (BCs) as early as possible [3]. The incubation and analysis of BCs are performed in microbiology laboratories. The shorter the period from drawing to final identification of the BC, the earlier the physician is able to change the therapeutic regimen from a broad to a small spectrum, from an effective to an optimal one.

During the last few years, several rapid diagnostic systems have become available for the identification of microorganisms in BCs [9,10]. One of these systems is the FilmArray^®^ Blood Culture Identification Panel (FA BCID) by BioFire/*bio*Mérieux^®^, a multiplex polymerase chain reaction (PCR) system capable of identifying 24 microorganisms and 3 antimicrobial resistance mechanisms in little over one hour. To date, studies have shown a significant time benefit by introducing the FA BCID to laboratory’s diagnostic pathways for BCs in regard to the time to identification and the time to optimal antimicrobial therapy in patients with BSI [11,12,13,14,15,16]. To this point, there has been no study investigating BSI by using FA BCID in a German hospital with only intensive care unit (ICU) patients. However, the impact of the FA BCID on the length of stay in the hospital or at the ICU itself, cost, and surrogate markers, e.g., procalcitonin (PCT) level [17], remains unclear. This study aims to prove the time benefit until an appropriate and effective antimicrobial therapy has been administered by introducing the FA BCID to the diagnostic pathway of BCs collected in the different ICUs of the Helios University Clinic Wuppertal, Germany. Furthermore, an earlier reduction of the PCT level, a reduction of the ICU length of stay as well as the mechanical ventilation time have been postulated.

Furthermore, certain subgroups of microorganisms are additionally analyzed. The duration of a systemic antimicrobial therapy in the case of the identification of only coagulase negative *Staphylococci* (CoNS) in positive blood culture vials (BC) is compared because, in many cases, these bacteria represent a contaminated BC and less true BSI cases. Therefore, it is postulated that antimicrobial therapy could be discontinued more rapidly in the intervention group.

Because the detection of *Staphylococcus aureus* in BC necessitates an immediate administration of appropriate antimicrobials, all the *S. aureus*-associated BSI episodes are of interest as well. The below stated endpoints are also applied for this subgroup analysis.

## 2. Methods and Materials

This study was conducted as an observational cohort study with a retrospective and a prospective cohort whose data were analyzed and compared. A randomization did not occur. Patient charts were electronically accessed to retrieve the necessary data. The study was approved by the ethics committee of the Witten/Herdecke University under the application number 166/2018. The prospective part took place in 2019–2020.

### 2.1. Objectives of the Study and Inclusion Criteria

The primary objective of this study was to determine the impact of the FA BCID in the laboratory routine on the time needed to identify microorganisms in positive BCs and on the time to an effective and appropriate antimicrobial therapy based on the FA BCID results and the recommendations of the sepsis guidelines. This study aims to investigate an improved patient care and outcome in BSI as well as sepsis with the hypothesis that adding the FA BCID as a diagnostic tool would shorten the time to an effective and appropriate antibiotic regimen compared to the retrospective control group. The patients were included int this investigational analysis if all criteria were fulfilled (Figure 1).

There were several secondary objectives and outcome measurements of this study. The markers to objectify the impact of the FA BCID compared to the retrospective group were the length of stay in the ICU, the time spent on invasive ventilation, and ICU mortality. The PCT level was used as a surrogate marker for the patient’s clinical response to the added FA BCID. A reduction of ≥80% from the first PCT level or <0.5 ng/mL was considered a successful therapeutic response to the administered regimen and the duration of time to achieve these targets was measured accordingly.

Patients at or above the age of 18 were eligible for inclusion if they had a BC taken at hospital admission or during their hospital stay that became positive while they were still alive at one of the ICUs of the Helios University Clinic Wuppertal, Germany. Written consent had to be obtained only for the prospective group.

### 2.2. Control Group Population and Conventional Identification Methods

The control group consisted of adult patients that were treated in either the surgical or the medical ICU or the intermediate care unit (IMC) at the Helios University Clinic Wuppertal during the time from April 2018 to December 2018. Blood cultures were processed as per laboratory directive. After extraction, the BCs were sent to the microbiology laboratory via transport personnel or an electronic transport system. In the laboratory, the BCs were placed inside the BACTEC™ system (Becton Dickinson, Heidelberg, Germany) and kept there for as long as five days. When the system sounded an alarm to inform the laboratory staff of growth in a BC vial, an aliquot of these cultures was taken, and a Gram stain was performed. Gram stain was routinely performed during the working hours of 7 am to 5 pm from Monday to Sunday batch-wise. The result of this Gram stain was then entered into the laboratory information system, which was then transferred to the hospital information system as soon as the physician in charge of the patient informed ICU staff of the result via telephone and fax. Depending on the Gram stain result, the positive BC was then sub-cultured on agar plates. The identification of the grown subcultures was performed via MALDI-TOF MS (matrix-assisted laser desorption ionization time-of-flight mass spectrometry; Bruker Daltonics, Bremen, Germany) and rarely using several validated desktop spot tests, such as catalase, oxidase, and agglutination tests. The automated system BD Phoenix™ (Becton Dickinson, Heidelberg, Germany) was utilized for the antimicrobial susceptibility testing (AST) of the grown subcultures according to the EUCAST guidelines (www.eucast.org accessed on 22 July 2020). AST was also carried out by disc diffusion tests. The physician in charge was informed of the results of the MALDI-TOF again via telephone and fax.

### 2.3. Intervention Group Population and Multiplex PCR FilmArray BCID Analysis

Two hundred positive blood cultures of BSI episodes represent the intervention group. Adult patients were recruited from the different ICUs (surgical, medical, and cardiothoracic-surgical ICUs and IMC) of the Helios University Clinic Wuppertal from January 2019 until July 2020. The collection of the BCs did not differ in the intervention group. The difference in handling of the BCs in the intervention group was the added step of the BioFire FilmArray^®^ Blood Culture Identification Panel (FA BCID) (BioFire Diagnostics, UT, USA) after a BC had become positive. The FA BCID is a rapid, multiplex PCR assay able to identify 24 different bacterial and fungal microorganisms as well as 3 resistance mechanisms within approximately 70 min. Targets not included in the FA BCID could not be analyzed by it and needed the routine laboratory workflow to identify the microorganisms. The results of the multiplex PCR were reported to the physician in charge via telephone and fax. Afterwards, the BCs were processed as in the control group to ensure the identification of all microorganisms, to perform the antimicrobial susceptibility testing, and to create a reference list for the FA BCID results. All data were collected from the two different groups. One group consisted of the individual patients.

### 2.4. Statistical Analysis

A Fisher exact test was used to compare the frequency distribution of categorical variables of independent groups.

For all metrical data, the data were checked for a normal distribution with a Shapiro–Wilk test. If a normal distribution was not identified, a Mann–Whitney U test was used to calculate the probabilities. A *t*-test was used if the data were distributed normally. The categorical data are presented with absolute and relative frequencies, whilst the metrical data with quantity, mean and medial value, standard deviation and the extreme values, i.e., minimum, maximum, and 25th and 75th percentile. A *p*-value of less than 0.05 was considered statistically significant. All tests were calculated two-sided. The *p*-values were interpreted in a descriptive manner.

## 3. Results

Overall, 329 individual patients with 364 BSI episodes were included in this FilmArray^®^ BCID sepsis study.

### 3.1. Patient Characteristics

Both groups consisted of more men than women, namely two thirds in the control group and 62.6% in the intervention group. The patients had a medial age of 68 years in both groups. No information was gathered on ethnicity, socioeconomic background, or comorbidities.

### 3.2. Primary Endpoints

Based on the results of BCID, the antimicrobial therapy (primary study endpoint) in BSI patients in the intervention group versus the control group could be adequately adjusted earlier in a median of 17 h (*p* = 0.071). While this does not show a statistically significant result, the amount of time saved is clinically important. Furthermore, we were able to show that changes in the antimicrobial therapy, which were not regarded as optimal therapy, but an improved therapy, nonetheless occurred, in a statistically significant manner, earlier in a median of 24 h (*p* = 0.029) (Table 1 and Table 2).

An effective and appropriate antimicrobial regimen in the subgroup of BSI caused by *Staphylococcus aureus* based on the BCID results was fulfilled earlier in the intervention group (*p* = 0.084) by a median of 18 h (Table 1 and Table 2). This again barely missed statistical significance.

### 3.3. Secondary Endpoints

The secondary endpoints of our study did not show significant results. It was not possible to conclude from the data that adding the FA BCID resulted in a shorter stay in the ICU, a shorter time spent on mechanical ventilation, a reduction in mortality, an earlier reduction of the PCT level of more than 80%, or a reduction in the duration of the antimicrobial treatment in general. Unexpectedly, the secondary endpoint of an earlier reduction of PCT level did occur in the control group (*p* = 0.003) (Table 1). Moreover, the duration of the antimicrobial regimen was not reduced in the intervention group with a BC revealing growth of only CoNS as compared to the control cohort (*p* = 0.041) (Table 3). Similar results were revealed for the duration of mechanical ventilation, which was shorter in this subgroup of the control group (*p* = 0.006) (Table 1 and Table 3).

The added FA BCID affected the antimicrobial regimen, in some way, in 22.2% of the cases in the intervention group. In 64.9% of the cases, no change occurred despite the use of the FA BCID. In the remainder 12.9% of the cases, the impact of the FA BCID could neither be determined as existent or non-existent.

We were able to report preliminary microbiology results nearly one day earlier compared to the usual preliminary results by the microbiology laboratory (Table 1).

Overall, the FA BCID demonstrated reliable results compared to the laboratory standard of care methods. The overall sensitivity of the FA BCID for all microorganisms was 92.82% with a specificity of 99.84% compared to the laboratory standard of care. Concerning only the on-panel microorganisms, the sensitivity was even higher with 98.48%.

## 4. Discussion

This study is the first of its kind to demonstrate a benefit of the use of the FA BCID in an ICU setting in Germany. We were able to show an earlier optimization of antimicrobial therapy in patients that were treated for BSI with the use of the FA BCID (Table 1 and Table 2). This result is comparable to prior studies that have been mostly performed in the United States. Banerjee et al. [11] and Bookstaver et al. [12] investigated the duration to de-escalation or escalation of the antimicrobial therapy, which is not necessarily the same as the duration to an optimal antimicrobial therapy. The former authors retrieved an earlier de-escalation of 13 h and an earlier escalation of 19 h, both of which come close to the 17 h found in our study. The latter authors found an earlier de-escalation for the FA BCID coupled with MALDI-TOF MS compared to MALDI-TOF MS alone by about 24 h for all antimicrobials and 12 h for anti-pseudomonal β-lactams. Again, these times are close to the results presented in this paper. MacVane and Nolte [13] found an earlier de-escalation rate by 14.9 h and also an earlier effective antimicrobial use by 10.1 h, the latter being defined as in this study. The study of Messacar et al. [14], whose patient population consisted of children, investigated the duration to optimal therapy as in our study and revealed a shorter duration to optimal therapy by 33.5 h. This may result from the fact that one third of the BCs were considered to be contaminated and an antimicrobial therapy was therefore already optimal when it had never been initiated. This, together with the unusually long time to the initiation of an antimicrobial therapy of about 48 h in the intervention group for the true BSI pathogens, might explain the big difference that has not been achieved in other trials. Buss et al. [18] achieved a shorter duration to an appropriate antimicrobial therapy through the FA BCID, which is an endpoint very comparable to our optimal antimicrobial therapy. It might even be a more realistic endpoint as the authors took the need for polymicrobial coverage into account, which was not performed in the definition used in our study. The authors found an improvement of duration to optimal therapy by 25 h. By using SeptiFast multiplex-PCR directly from blood samples, Plettig et al. identified the causal pathogens 50 h earlier than the standard culture method [19].

Our primary endpoint did not allow an antimicrobial regimen to be optimal if the patient was already on broad-spectrum antibiotics due to another infectious focus with a different pathogen than the one in the BC. This fact may have confounded our results in a way that the impact of the FA BCID might even be more considerable. This makes the result of the shorter duration until a change in the antimicrobial therapy occurs so interesting. A median of 24 h earlier change of the antimicrobial therapy is about the same time that the identification process was accelerated by the FA BCID. The statistical significance of this result (*p* = 0.029), in contrast to the primary endpoint, highlights the importance.

Even though previous studies postulated the effect on patient outcome in regard to the length of stay, mortality, or length of ventilation [11,13,15,20,21], we were unable to assess any potential beneficial outcomes reading these aspects. The reason for this can be that it is possible that the effect of the earlier optimization is not high enough to affect the other parameters/variables. Furthermore, the clinical presentation of ICU patients with BSI is extremely complex and an antimicrobial therapy is only one of many adjustable variables. Most patients received a very broad antimicrobial coverage to begin with, so that optimization usually meant de-escalation rather than escalation. Undertreating a patient is usually more harmful than overtreatment, especially in the setting of BSI and sepsis, which could further explain the missing impact. The low incidence of multidrug-resistant bacteria at the Helios University Clinic Wuppertal (five ESBL, three ciprofloxacin-resistant ESBL-producing isolates, one carbapenemase-producing isolate, and one MRSA isolate in the prospective study group) also simplifies the antibiotic treatment and prevents suboptimal treatment, and therefore results in longer stays in the ICU and higher mortality rates.

Nonetheless, we found that, in 22.2% of the cases, the antimicrobial regimens were adjusted according to the FA BCID results. However, this is much lower than what was revealed by Ray et al. [16], who reported 54% adjustments.

Patient recruitment for the intervention group was subject to a high selection bias that was caused by the requirements made by the ethics committee to obtain written consent from each patient. Most of ICU patients were incapacitated and therefore unable to give consent to participate in the study. Often a legal guardian was not yet established or could not be reached in a timely manner. As these patients represent a sicker portion of ICU patients, they might be under-represented in the intervention group. No restrictions were in place for inclusion in the control group and, therefore, this patient collective might be much more diverse.

### 4.1. Subgroup of Coagulase Negative Staphylococci

Blood-culture vials may be contaminated due to erroneous specimen collection processing. Typical contaminants include skin flora, such as coagulase-negative *Staphylococci* (CoNS), *Streptococcus viridans*, *Cutibacterium*, *Corynebacterium*, and *Bacillus* spp. The impact of contaminated BC can be significant, leading to unnecessary antimicrobial treatment and diagnostic procedures and extended hospital stays, with average costs per episode of USD 8000–25,000 [22]. The comparison of the control and intervention groups concerning those BCs resulting in only CoNS brought several more unexpected results. Comparability was somewhat limited as the mortality in the control group is 6% higher. The number of BSI episodes available for further analysis compromised the quality of the data. While the overall case numbers in the control group were smaller, the number of CoNS–BSI cases was higher than in the intervention group (by approx. 23%), which may lead to the intervention group being underpowered. Whether a CoNS result was considered true or as a contaminant, the decision was taken by the ICU physicians according to the clinical situation of the patient, which could lead to even further bias due to personnel changes over the study period. This subgroup analysis was conducted because we had anticipated a shorter duration of antimicrobial therapy in the intervention group. The results show the opposite is true for our study cohort (Table 3). The total duration of antimicrobial therapy was significantly shorter in the control group (*p* = 0.041). Unfortunately, other clinical trials did not conduct this subgroup analysis. They also did not compare the duration of mechanical ventilation for CoNS cases. This was performed in our study as well. The duration of mechanical ventilation was significantly shorter again in the control group compared to the intervention group (*p* = 0.006). Moreover, the number of BSI cases in the control group was much higher, by around 28%. A possible explanation for these two unexpected results might be that BSI patients with CoNS in the intervention group had more severe events and were not of need for further antimicrobial therapy. We believe it is unlikely that the FA BCID results per se did cause the prolonged duration of mechanical ventilation. 

As the FA BCID 2.0 panel includes the highly prevalent *Staphylococcus epidermidis* to the species level, it would be interesting to see a trial conducted with the newer panel examining BSI cases caused by CoNS. Even though the results of this study contradicted our hypothesis (Table 3), it is still possible that FA BCID 2.0 is able to reduce the duration of unnecessary antimicrobials in BSI cases caused by CoNS.

### 4.2. Subgroup of Staphylococcus aureus

As postulated, the time to an optimal antimicrobial therapy in the BSI cases caused by *S. aureus* was shorter in the intervention group than in the control group (*p* = 0.081). Again, this endpoint barely misses statistical significance, likely because of this study group being underpowered. Nonetheless, we consider this time saving clinically important. The result is easily explainable by the earlier identification with the FA BCID compared to the laboratory standard of care. The identification took a median of 23 h from the drawing of the BC to the communication of the FA BCID result. The optimization took a median of 22 h, suggesting an immediate action on the results. In one case, the FA BCID result led to an earlier removal of an implanted port system that had been the source of the *S*. *aureus* BSI. These results underline the special importance of rapid diagnostic tools, such as the FA BCID, in cases where certain bacteria are the cause of a BSI or sepsis. It is unlikely that an even earlier optimization is possible at this time, because the process of transporting the BC to the laboratory and incubating it takes up most time. Only a direct identification from whole blood would speed up the process even further.

Similar to previous studies [23,24,25,26,27,28,29], we were also able to confirm the reliability of the FA BCID results compared to the laboratory standard of care.

### 4.3. Further Findings

In our sepsis study, we had only three fungal BSI cases. Invasive candidiasis ranges from 5 to 10 cases per 1000 ICU admissions and represents 5% to 10% of all ICU-acquired infections, with an overall mortality comparable to that of severe sepsis/septic shock. In Europe, *C. albicans* remains the most frequent species, and epidemiological trends suggest that non-albicans Candida species, in particular *C. glabrata*, are emerging [30]. In critically ill patients, especially in those with high-risk scores and empirical therapy in septic patients not responding to appropriate antibacterial treatment, caspofungin is the antifungal of choice according to the guidelines. However, FA BCID would not lead to de-escalation, e.g., from caspofungin to fluconazole if *Candida albicans* was identified. Moreover, PCT level determination is of no benefit in candidemia. In fungal BSI, one should follow the recommendations of the guidelines for the duration of antifungal regimen. Charles et al. conducted a retrospective study enrolling 50 nonsurgical septic patients with 35 bacteremia and 15 candidemia cases, and revealed a significantly lower PCT level in patients with candidemia (median 0.65 ng/mL) compared to those with bacteremia (median 9.75 ng/mL) [31]. A prospective study of Martini et al. revealed similar results for candidemia cases: PCT levels were lower in patients with candidemia (median 0.71 ng/mL) than in those with bacteremia (median 12.9 ng/mL) [32].

A limitation of our study is that it was performed at the ICU department, so it may not be generalizable to other wards or clinics. Other improvements, such as educational courses for antimicrobial stewardship that were taking place, may have affected our study results.

## 5. Conclusions

The appropriate antimicrobial regimen and implementation of the FA BCID in patients with severe sepsis and septic shock due to bloodstream infection was associated with improved patient care by optimizing an antimicrobial regimen earlier in BSI patients. Surrogate markers indicating an earlier recovery of the BSI patients of the intervention group, such as the length of stay at the ICU, duration of mechanical ventilation, or a more rapid reduction of PCT level, were not significantly affected. However, mortality rate did not differ between the two patient groups (control versus intervention). A presumed de-escalation or stop of the antimicrobial therapy in those cases in which CoNS were identified did occur in the control group, but not in the intervention group (*p* = 0.041). Further investigation of BSI should be considered by use of the updated FA BCID 2.0 panel with more gene targets to assess impacts on patient survival, length of stay, and length of mechanical ventilation, and response to de-/escalation of the antimicrobial regimen.

## Figures and Tables

**Figure 1 jcm-10-05880-f001:**
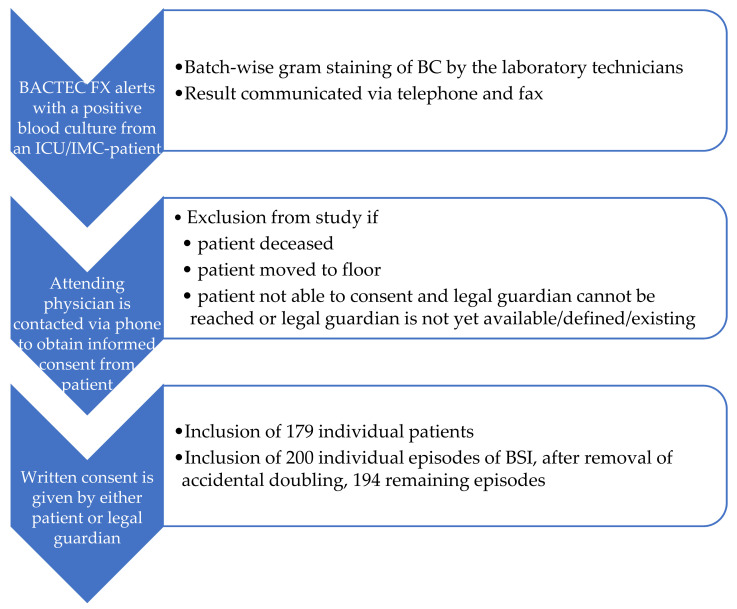
Diagram of the patient inclusion process into the intervention group. ICU, Intensive Care Units; IMC, intermediate care unit; BC, blood culture.

**Table 1 jcm-10-05880-t001:** Overview of the statistical results obtained via the comparison of the variables of the two study groups (control versus intervention).

Variable	C/I	*n*	Mean	SD	Min	25th Perc.	Median	75th Perc.	Max	*p*-Value *
Duration until antimicrobial change (h) ^1^	C I	71 66	167.6 92.9	447.2 181.4	0 0	34 15	60 36	120 94	3180 1227	0.029
Time to optimal therapy (h) ^1^	C I	101 118	41.3 25.4	48.3 29.3	0 0	0 0	37 20	63 41	242 131	0.071
Duration of antimicrobials (h) ^2^	C I	137 157	209.0 212.0	363.0 290.5	2 1	77 58	120 136	199 253	3300 2615	0.986
Duration	C	149	305.2	410.3	20	99	187	343	3329	0.241
of ICU stay (h) ^2^	I	178	312.7	412.8	10	65	164	406	2871	
Duration of ventilation (h) ^2^	C I	83 87	428.4 311.5	1938.6 453.2	6 2	30 52	104 154	236 406	17,616 2783	0.109
Time to 80% PCT level reduction (h) ^2^	C I	49 44	124.1 190.4	76.9 138.8	15 50	70 113	113 147	168 211	345 716	0.003
Time to optimal therapy (h) ^3^	C I	14 12	51.3 30.0	41.6 28.3	0 0	23 21	40 22	94 31	122 106	0.084

^1^ All BSI episodes; ^2^ Individual patients with all microorganisms; ^3^ Only *Staphylococcus aureus* BSI episodes. C = control group; I = intervention group; * *p*-value of Mann–Whitney *U* tests; ICU, Intensive Care Units; PCT, procalcitonin; SD, standard deviation

**Table 2 jcm-10-05880-t002:** Overview of study endpoints.

**Primary endpoint**	Time to effective/appropriate antimicrobial therapy ^1^ In subgroup of *Staphylococcus aureus* BSI	rejected: *p* = 0.071 rejected: *p* = 0.084
**Secondary endpoints**	ICU length of stay	rejected: *p* = 0.241
	Duration of mechanical ventilation	rejected: *p* = 0.109
	Mortality	rejected: *p* = 0.135
	PCT level reduction of ≥80% or <0.5 ng/mL	rejected, faster drop-in control group, *p* = 0.003
	Duration of systemic antimicrobial therapy in CoNS-cases	rejected, shorter duration in control group, *p* = 0.041
**Others**	No effect in BSI cases caused by fungi	

^1^ Based on FilmArray BCID results and/or international/national guidelines.

**Table 3 jcm-10-05880-t003:** Overview of statistical results in BSI caused by *CoNS*.

Variable	C/I	*n*	Mean	SD	Min	25th Perc.	Median	75th Perc.	Max	*p*-Value *
Duration	C	59	160.9	151.3	2	65	117	199	765	0.041
of antibiotics (h)	I	48	256.0	259.6	3	80	166	362	1247	
Duration	C	50	206.2	327.9	6	23	94	236	2073	0.006
of ventilation (h)	I	39	340.9	319.1	6	119	268	519	1296	
Duration until 80 % PCT level reduction (h)	C I	20 12	120.9 204.7	83.6 128.0	44 68	70 116	107 179	134 259	345 505	0.015

Only individual patients, *Staphylococcus* spp. other than *Staphylococcus aureus.* C = control group; I = intervention group; * *p*-value of Mann–Whitney U tests.

## Data Availability

The datasets used and analyzed during the current study are available from the corresponding author on reasonable request.

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
