# Peer review of "The Impact of the FilmArray-Based Detection of Microbial Pathogens from Positive Blood Culture Vials on the Time to Optimal Antimicrobial Regimen in Intensive Care Units of the Helios University Clinic Wuppertal, Germany"

_jcm, 2021, doi:10.3390/jcm10245880_

Round 1

Reviewer 1 Report

My comment is concerned with the author can provide more information about the FA BCID in the introduction part to make it clear about the advantage or the disadvantage of the FA BCID. The overall methodology is well organized 

Author Response

My comment is concerned with the author can provide more information about the FA BCID in the introduction part to make it clear about the advantage or the disadvantage of the FA BCID. The overall methodology is well organized 

                             Schumann: Disadvantages included in lines 149 - 151

Reviewer 2 Report

The study lacks some originality as the role of rapid diagnostic tests has been previously investigated and has several limitations. Thus, it does not add significant knowledge improvement. Moreover, my principal concern is about statistical analysis and interpretation of results. In my opinion, the authors' interpretation is not pertinent to the results as the p-value considered for significance was 0.1. Moreover, the inclusion and exclusion criteria are not clear. Finally, the result section does not report the patients' characteristics.

Minor comments:

  • Table 1 does not seem adequate to the context where it is cited.
  • Tables should report mean (sd) or median (IQR) according to data distribution.

Author Response

The study lacks some originality as the role of rapid diagnostic tests has been previously investigated and has several limitations. Thus, it does not add significant knowledge improvement. Moreover, my principal concern is about statistical analysis and interpretation of results. In my opinion, the authors' interpretation is not pertinent to the results as the p-value considered for significance was 0.1. Moreover, the inclusion and exclusion criteria are not clear. Finally, the result section does not report the patients' characteristics.

                             Schumann: p-value has been reconsidered and changed to 0.05. Inclusion criteria have been added (lines 114 – 117). Patient characteristics have been added, lines 170 - 174

Minor comments:

  • Table 1 does not seem adequate to the context where it is cited.
  • Tables should report mean (sd) or median (IQR) according to data distribution.

Schumann: Table 1 removed, others have been renamed accordingly. Mean and median data distribution are included in tables 1 and 2

Reviewer 3 Report

Jannik Schumann et all in this study provide very important data that could be useful for bloodstream infection pathogen earlier identification and management. This manuscript investigated an original research question that would improve antimicrobial prescription and rational use.

However, the following major comments and suggestions would improve the manuscript quality.

Background

  1. Table 1 could be deleted the sentence describing the contents of the table and the reference is sufficient

Study design and inclusion criteria session

  1. The text in this session was related to the objectives of the study. Please change the session title or complete the text to include the design of the study
  2. “The patients were included in to this investigational analysis if all criteria were fulfilled “     it should be "into"
  3. What are the inclusion criteria? this should be added in the text
  4. What was the study period?
  5. Study population: please describe more the study population (adult or children, was the blood collected at admission or during the hospitalization) the reader may be interested in the profile of the study population
  6. Study design: My major concern is about the study design.

Am not sure that this study is interventional.

 What is the intervention?

  How the intervention group is selected?

 Is the participant randomized?

  This study seems to be observational than interventional since you compared two groups with two different routine diagnosis procedures.

I would suggest the author consider the study as an observational cohort study with retrospective and prospective data analysis

Result

  1. Line 182: Thesis or study?
  2. Line206: The studies mentioned were performed where?
  3. Minor comment: Consider revising and reducing the list of keywords

Author Response

Jannik Schumann et all in this study provide very important data that could be useful for bloodstream infection pathogen earlier identification and management. This manuscript investigated an original research question that would improve antimicrobial prescription and rational use.

However, the following major comments and suggestions would improve the manuscript quality.

Background

  1. Table 1 could be deleted the sentence describing the contents of the table and the reference is sufficient

Schumann: Table 1 has been deleted

Study design and inclusion criteria session

  1. The text in this session was related to the objectives of the study. Please change the session title or complete the text to include the design of the study

Schumann: Changed the titles (lines 95, 118, 140), design has been added line 89 – 90.

  1. “The patients were included in to this investigational analysis if all criteria were fulfilled “     it should be "into"

Schumann: changed, see line 103

  1. What are the inclusion criteria? this should be added in the text

Schumann: has been added, lines 114-117

  1. What was the study period?

Schumann: added, line 94 and 143/144

  1. Study population: please describe more the study population (adult or children, was the blood collected at admission or during the hospitalization) the reader may be interested in the profile of the study population

Schumann: population described more detailed, lines 113-116, 119, 142

  1. Study design: My major concern is about the study design.

Am not sure that this study is interventional.

 What is the intervention?

  How the intervention group is selected?

 Is the participant randomized?

                             Schumann: No randomization, addressed in line 90

  This study seems to be observational than interventional since you compared two groups with two different routine diagnosis procedures.

I would suggest the author consider the study as an observational cohort study with retrospective and prospective data analysis

                             Schumann: Study design has been changed: lines 89-90

Result

  1. Line 182: Thesis or study?

Schumann: changed, line 196

  1. Line206: The studies mentioned were performed where?

Schumann: location added, line 220-221

  1. Minor comment: Consider revising and reducing the list of keywords

Schumann: Removed MALDI-TOF

Round 2

Reviewer 2 Report

Thanks to the authors for the revised version of the manuscript that presents some improvements. However, I still have concerns about data robustness and interpretation. The term "numerically significant" does not seem appropriate. The authors give too much emphasis to the negative results, presenting them as "numerically significant". Finally, patients' characteristics are not extensively reported.

Author Response

Thank you very much for the second review and the detailed criticism of our manuscript (Manuscript ID: jcm-1462794) with the title “Impact of the FilmArray based detection of microbial pathogens from positive blood culture vials on the time to optimal antimicrobial regimen in intensive care units of the Helios University Clinic Wuppertal, Germany” by Schumann et al. The following is intended to give a better understanding of the adjustments made according to the remaining reviewer’s concerns.

We agree that the term “numerically significant” is misleading. Therefore, it was removed throughout the manuscript and especially from table 3. Lines 175 – 185 have been adjusted accordingly as labeled in the comments.

In lines 179 – 180 the statistically significant result has been clarified to put more emphasize on it.

The subgroup results for S. aureus have been modified in lines 313 – 315. The emphasize on the numerically relevance is no more.

Patient characteristics are reported as far as possible. No more characteristics have been investigated, this has been addressed in lines 170 – 173.

We added another subsection 4.3. for better readability of the discussion.

Again, minor typos and misleading words have been changed and labeled accordingly.

Reviewer 3 Report

This is fine.

The manuscript has been improved.

I have not further comment 

Author Response

Thank you very much for the second review.